# Flow-sensory contact electrification of graphene

Xiaoyu Zhang [1], Eric Chia [1,3], Xiao Fan[1,3] & Jinglei Ping [1,2]✉

All-electronic interrogation of biofluid flow velocity by electrical nanosensors incorporated in ultra-low-power or self-sustained systems offers the promise of enabling multifarious emerging research and applications. However, existing nano-based electrical flow sensing technologies remain lacking in precision and stability and are typically only applicable to simple aqueous solutions or liquid/gas dual-phase mixtures, making them unsuitable for monitoring low-flow (~micrometer/second) yet important characteristics of continuous biofluids (such as hemorheological behaviors in microcirculation). Here, we show that monolayer-graphene single microelectrodes harvesting charge from continuous aqueous flow provide an effective flow sensing strategy that delivers key performance metrics orders of magnitude higher than other electrical approaches. In particular, over six-months stability and sub-micrometer/second resolution in real-time quantification of whole-blood flows with multiscale amplitude-temporal characteristics are obtained in a microfluidic chip.

---

[1] Department of Mechanical and Industrial Engineering, University of Massachusetts Amherst, Amherst, MA, USA. [2] Institute for Applied Life Sciences, University of Massachusetts Amherst, Amherst, MA, USA. [3] These authors contributed equally: Eric Chia, Xiao Fan. ✉email: ping@engin.umass.edu

Electrical transducers based on nanomaterials hold great potential for interrogating biofluid-flow velocity in self-powered or ultra-low-power systems[1–6]. An electrokinetic approach[3–5] is to use nanotransistor devices to measure flow-dependent streaming potential/current. The devices are miniaturized but subject to an intrinsic resolution limit of ~80 µm s$^{-1}$ (in a 1-Hz bandwidth) induced by thermal noise and liable to be reduced to ~20 mm s$^{-1}$ in a non-ideal measurement system. Alternatively, triboelectric charge harvested from a liquid flow by a micro/nanoelectrode device can be quantified for gauging the flow velocity. However, existing flowmeters based on this strategy typically use large-size (>mm$^3$) bundled nanotube/nanowire transducers[6] that are difficult to be scaled down, easy to cause flow-channel clogging, and prone to signal weakening and distortion due to the fouling of the electrodes upon specific/non-specific electrochemical processes and physicochemical adsorptions[7,8]. Recently, flow sensors enabled by the cyclical formation of the electrical double layer (EDL) of the aqueous solution at a solid-aqueous interface have been developed but they are only suitable for liquid/gas dual-phase mixtures (cavity-confined solution[9,10], droplets[11], and waving water[12]).

Here, we show self-powered graphene microdevices that transduce in real time the flow of continuous blood in a microfluidic channel to charge-transfer current in response to the flow-sensory rearrangement (not formation/deformation) of EDL at the graphene-aqueous interface. The devices deliver a resolution of $0.49 \pm 0.01$ µm s$^{-1}$ (in a 1-Hz bandwidth), a two-orders-of-magnitude improvement compared with existing device-based flow-sensing approaches, and are ultrathin (one-atom-layer) and of low risk of being fouled or causing channel clogging. For periods exceeding 6 months the devices have demonstrated minimal variations in key performance metrics.

## Results

**Graphene–water contact electrification.** The flow transduction of the devices is based on a single microelectrode of monolayer graphene that harvests charge from flowing blood through contact electrification without the need for an external current supply. For implementing blood-flow measurements, we fabricated acrylic chips with a graphene single-microelectrode device

extending over the microfluidic channel (Fig. 1a). The monolayer graphene was prepared via chemical vapor deposition (CVD) and transferred to the chip through a low-contamination electrolysis method[13]. The flow pathway, including the microfluidic chip and the tubes that connect to the chip, is entirely based on electro-chemical inert materials. A flow of EDTA-anticoagulated whole bovine blood (pH = 7.0, ionic strength = 150 mM) with precisely controlled velocity was driven through the microfluidic channel by a syringe pump. The graphene microelectrode was wired to the inverting input of an operational amplifier of a coulombmeter. The charge harvested from the solution by the graphene was stored in a feedback capacitor of the amplifier and quantified (Fig. 1a).

The charge transferred into a graphene device was measured as a function of time at various blood-flow velocities. For each flow velocity, the transferred charge (Fig. 1b) indicates a clear, robust proportional relationship with respect to time. The charge-transfer current, obtained by proportional fit to the charge–time data, is of high precision (<10 fA) and signal-to-noise ratio (49.2 ± 0.7 dB), enabled by the low-noise characteristics of electron transfer at the interface of graphene with aqueous solution. The current can be smoothed by using a Savitzky–Golay filter with controllable bandwidth. The level of the current is minimally associated with the number of the graphene edge states (Supplementary Fig. 1). According to the electron-transfer mechanism for solid/water contact electrification[14], the charge transfer occurs via quantum tunneling of electrons through the electronic states of the defects (dangling bonds, electrical disorders, and grain boundaries) on the graphene basal plane[15–17] as demonstrated in Supplementary Fig. 2, entailing the ultra-low current noise level intrinsic to graphene-aqueous interfaces. Provided that the electrical conductance corresponding to a typical electron-tunneling process is ~$10^{-24}$ $\Omega^{-1}$ [14], the specific charge-transfer conductance of as-prepared CVD graphene ($\sigma_{ct}$), approximately equal to ~$1.5 \times 10^{-9}$ $\Omega^{-1}$ mm$^{-2}$ as measured in our previous work[18], leads to a defect density of ~$10^{17}$ cm$^{-2}$. This value agrees with that obtained from Raman spectroscopy of the graphene prepared, $1.44 \times 10^{17}$ cm$^{-2}$ [18], well in line with the quantum-tunneling mechanism of the charge-transfer current at graphene-defect states.

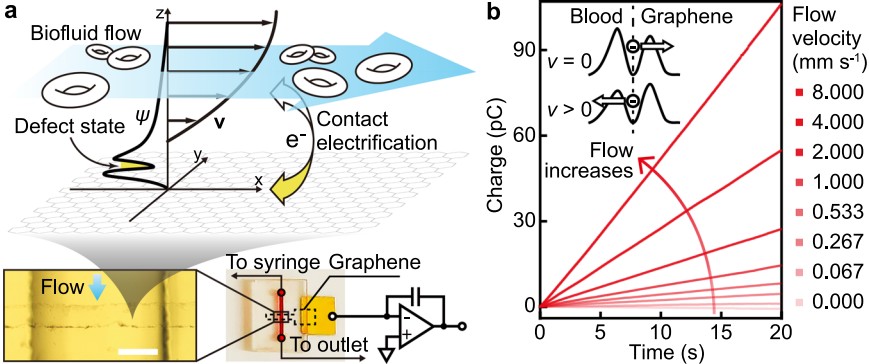

**Fig. 1 Transducing blood-flow velocity to electric current by using a graphene single-microelectrode device. a** Coulometric measurement of contact-electrification charge transfer between whole-blood flow and graphene. Graphene is shown by the gray honeycomb lattice. The whole blood contains multifarious components such as red cells which are depicted in the schematic. The direction of the whole-blood flow is represented by the blue arrow. The thick curves represent the profiles for the flow velocity (**v**) and the electrical potential ($\psi$) along the z-axis normal to the graphene–water interface (x–y plane). The flow velocity field is shown by the thin black arrows enveloped by the velocity profile. The charge transfer indicated by the white-yellow arrow is through the graphene-defect state highlighted on the $\psi$ profile. The optical microscope image shows a monolayer-graphene microelectrode crossing over the microfluidic channel in an acrylic chip. The scale bar is 200 µm. The graphene microelectrode is connected to the gold contact that is wired to an electrometer based on an operational amplifier with a feedback capacitor. **b** The measured unsmoothed charge transfer of a graphene device as a function of time for different blood-flow velocities. The diagram illustrates the electrical potential variation at a graphene-defect site and the corresponding variation in the electron transfer through the defect state, when the blood-flow velocity magnitude (v) changes. The vertical dash line indicates the graphene-defect state at the graphene/blood interface. The arrows represent the directions of the net electron flow.

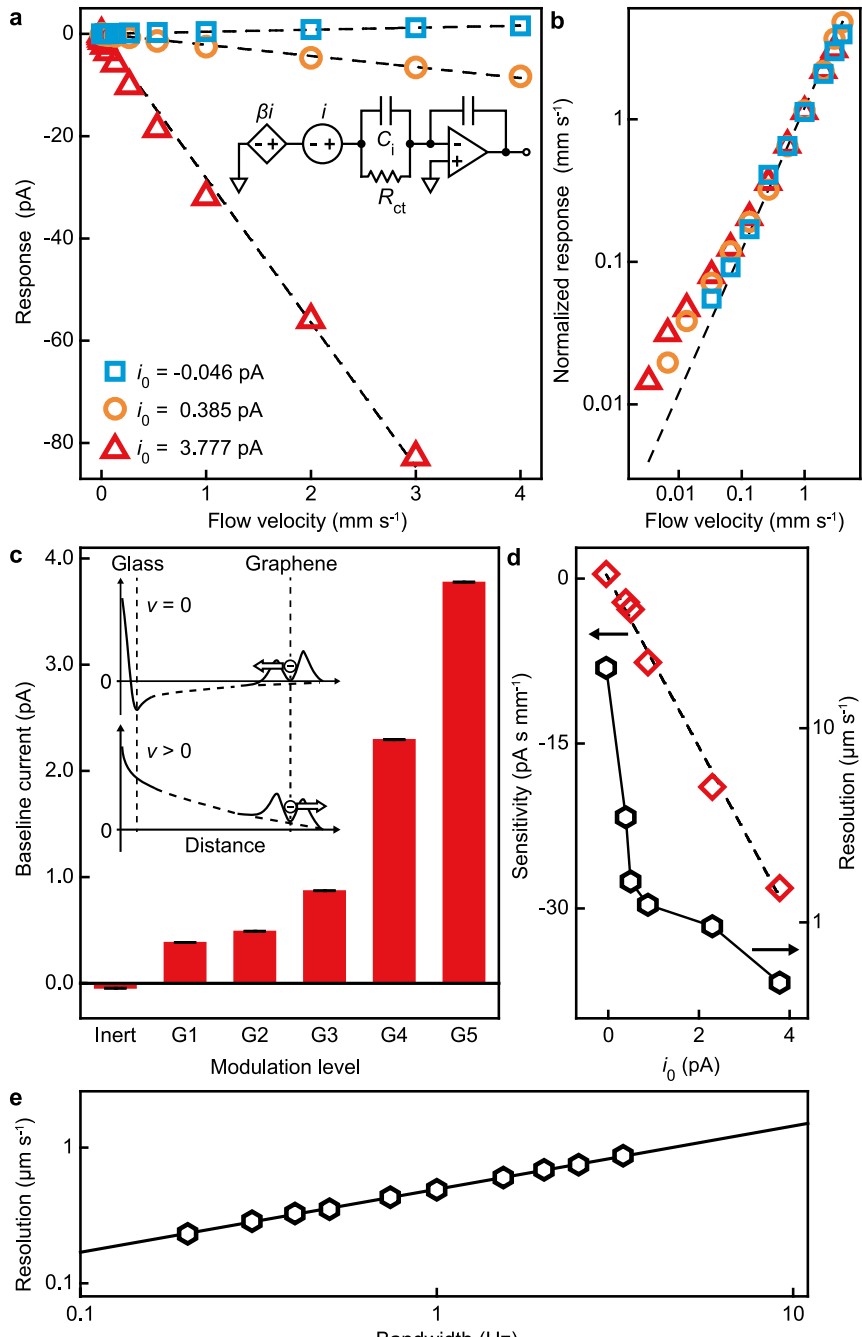

**Fig. 2 Response curves and characteristics for blood-flow-velocity quantification by the graphene single-microelectrode device. a** The current response as a function of flow velocity. In the circuit, $\beta$ is the scaling factor that models the modulation of the graphene charge-transfer current ($i$) by an unbiased non-faradaic electrode made of glass with variable surface charge density and $i_0$ is the baseline current (the current at zero flow velocity). The dash lines are best proportional fits to the data. The linear electrical circuit models the charge-transfer current through the graphene/blood interface represented by a charge-transfer resistance ($R_{ct} = 1/\sigma_{ct}$) and an interfacial capacitance ($C_i$). **b** The sensitivity-normalized current response as a function of flow velocity. The eye-guiding dash line is of the unit slope. The data symbols are the same as in (**a**). **c** The baseline current (at zero flow velocity) measured with (G1–5) and without (Inert) using an unbiased glass electrode. The energy profile diagram illustrates the modulating effect of the charged glass electrodes on the graphene–blood electron transfer through a graphene-defect electronic state when the magnitude of the flow velocity ($v$) increases. The vertical dashed line on the glass (graphene) side represents the glass/blood (graphene/blood) interface. The error bars are charge-transfer current standard deviations in 1-Hz bandwidth. **d** Resolution and sensitivity as a function of the baseline current $i_0$. The dash line is a proportional fit to the sensitivity data. **e** Resolution as a function of bandwidth for $i_0$ equal to 3.777 pA (G5). The black line is based on the parameters of the best linear fit to the data. In **a**, **b**, **d**, and **e**, the sizes of the error bars (the standard deviations of corresponding quantities based on the current uncertainties over the measurement bandwidth (1 Hz for **a**, **b**, and **d**)) are smaller than the size of the data points.

**Linear current–flow relationship**. As shown by Supplementary Fig. 3, the charge-transfer current of the graphene device is linearly associated with the blood-flow velocity, leading to a proportional relationship between the current response (the flow-induced current variation relative to the current at zero flow velocity) and the flow velocity (Fig. 2a, blue squares). The linear relationship covers over four orders of magnitude of flow velocity and is well reflected by the unit-slope power law of the current response normalized by the (global) sensitivity (the slope of the best proportional fit to the current–velocity data) as a function of flow velocity (Fig. 2b, blue squares).

Our control experiments indicate that the linear current–velocity relationship is not induced by piezoelectric effects (Supplementary Fig. 4) or charge transfer at the graphene edge electronic states (Supplementary Fig. 1). The effect of the streaming potential from the syringe to the graphene single microelectrode on the charge-transfer current at the graphene microelectrode is also negligible. First, the streaming potential depends on the length of the blood remained in the syringe during the measurement while in our experiment the measured current at the graphene microelectrode was independent of the length. Furthermore, the streaming potential is <2.7 µV according to $\Delta V \approx (\varepsilon \zeta \Delta p / \sigma \eta)$, where $\varepsilon$ is the blood permittivity ($7.68 \times 8.854 \times 10^{-12}$ F m$^{-1}$), $\zeta$ the zeta potential (<60 mV in magnitude), $\Delta p$ the pressure drop (1650 Pa), $\sigma$ the blood electrical conductivity (~1.23 S m$^{-1}$), and $\eta$ the blood viscosity (~3.5 cP)[19]. Adding this potential to the electrical potential at the graphene EDL generates negligible, <2.0-fA charge-transfer current at the graphene (provided that the charge-transfer resistance, $R_{ct}$, is ~1.3 GΩ)[18]. Considering the electron-tunneling origin of the graphene–water contact-electrification current, the flow transduction can be understood by the flow-induced rearrangement of the EDL at the graphene basal-plane defects: An increase in the blood-flow velocity increases the wall shear stress generated by the flow at the graphene-aqueous interface (Fig. 1a), suppresses the counterions' screening effect[20], and reduces the electrical potential barrier at the blood side of the graphene-defect electronic states, resulting in enhancement in the transferring of electrons from the graphene to the blood, as shown by the energy profile in Fig. 1b. This EDL-rearrangement mechanism represents a flow-sensory modality different from the typical mechanism that is based on the formation/deformation of EDL (or moving EDL boundary) at the solid–liquid interface in previous prominent studies[9–12,21]. As the blood-flow increases in velocity, the EDL rearrangement reverses the polarity of the current and then increases its magnitude (Supplementary Fig. 3)[22]. To the first order of magnitude, the impact of flow velocity $v$ on the charge-transfer current $i$ can be written as $i \approx i_0 + sv$, where $i_0$ is the baseline current (the current at zero flow velocity) and $s$ is the sensitivity.

The first-order EDL-rearrangement scenario for the flow-sensory charge-transfer current of the graphene devices suggests that the sensitivity of the device can be improved by modulating the electrical potential distribution of the EDL at the graphene/blood interface to enhance the charge-transfer current thereat. Biased non-faradaic counter electrodes made of inert metals, such as platinum, are typically unstable and can generate substantial (>10–100 pA) background noise to the flow-induced charge-transfer current signal measured by the graphene microelectrode, thereby burying the signal in the noise. In order to control the charge-transfer current at an accuracy of sub-pA level, we used an unbiased non-faradaic electrode made of glass with variable surface charge density that can generate correspondingly different ultra-stable electrical potential coupling to the graphene EDL potential structure. As shown in the energy profile diagram in Fig. 2c, at $v = 0$, the adsorption of specific counterions renders the electrostatic potential reversed at the glass EDL[23–25] and reduces the electrical potential barriers on both the blood side and the graphene side at a graphene disorder electronic state to different extents. The result is that the potential barrier on the graphene side is relatively higher than that on the blood side and more electrons transfer out from the graphene to the blood compared with the situation without the modulation. In comparison with newly used glass, the surface charge density of aged glass is higher due to the enhancement of the ionization effect at the glass surface, leading to greater change in the energy profile and, in consequence, more enhanced baseline charge-transfer current of the graphene devices (Fig. 2c). The level of the baseline current (~pA) leads to specific charge-transfer conductance of ~$10^{-9}$ Ω$^{-1}$ mm$^{-2}$, well agreeing with ~$1.5 \times 10^{-9}$ Ω$^{-1}$ mm$^{-2}$ in our previously published work using SiO$_2$/Si substrate with steady charging state (electrostatic potential) that modulated the graphene charge-transfer current[18].

Figure 2a, b and Supplementary Fig. 5 show that at each modulating level represented by a corresponding baseline current, the current response of the graphene device is proportional to the flow velocity, as in the unmodulated experiment, aside from minimal deviations at flow velocities <0.1 mm s$^{-1}$ induced by the modulation. An enhanced charging status (higher baseline current magnitude) of the glass electrode corresponds to higher magnitude of the charge-transfer current response of the graphene device and higher magnitude of the slope of the current–velocity data, i.e., higher sensitivity for flow velocity determination (Fig. 2d). The sensitivity enhancement delivers an optimal resolution (in a 1-Hz bandwidth) of $0.49 \pm 0.01$ µm s$^{-1}$ (Fig. 2d and Supplementary Fig. 6), a two-orders-of-magnitude improvement compared with previous device-based flow sensors (Supplementary Table 1). The remarkable resolution of the flow-velocity measurement is imparted by the low-noise electron transfer of graphene in contact with the blood flow. The log–log plot of the flow-velocity resolution vs. bandwidth relationship of our measurement (Fig. 2e) shows a clear power law. The slope of the best linear fit is $0.47 \pm 0.01$, suggesting a square root dependency of the current noise with bandwidth, in good agreement with the thermal noise arising from the charge-transfer resistance at the graphene–water interface: $i = \sqrt{4k_B T \Delta f / R_{ct}} = 3.56 \sqrt{\Delta f}$ (fA Hz$^{-1/2}$), where $k_B$ is the Boltzmann constant, $T$ the temperature (298 K), and $\Delta f$ the bandwidth.

Under the first-order EDL-rearrangement scenario, the graphene charge-transfer current modulation can be modeled by adding a dependent current source (with a scaling factor $\beta$) to the linear circuit description of the graphene/blood interface (Fig. 2a): $i \rightarrow i + \beta i$. Considering the current–flow relationship ($i = i_0 + sv$), $i_0$ and $s$ are multiplied by the same factor: $i_0 \rightarrow (1 + \beta)i_0$ and $s \rightarrow (1 + \beta)s$, yielding a proportional $i_0$–$s$ relationship over the whole range of $i_0$, in good agreement with Fig. 2d. The slope ($0.130 \pm 0.004$ mm s$^{-1}$) of the best proportional fit represents a common blood-flow velocity corresponding to zero charge-transfer current for all modulation levels. This property is useful for device calibration when the modulation is at an unknown level.

We also measured the contact-electrification charge transfer for the flow of phosphate-buffered saline (PBS) at different velocities by using the same graphene device that was used in the blood-flow measurement. The transferred charge shows clear linearity with respect to time for each flow velocity and the current response is proportional to the flow velocity (Supplementary Fig. 7), as in the blood-based experiment. The magnitude of the sensitivity of the device for PBS velocity determination is 4.6× smaller than that of whole blood due to the lower viscosity of PBS that generates lower wall shear stress

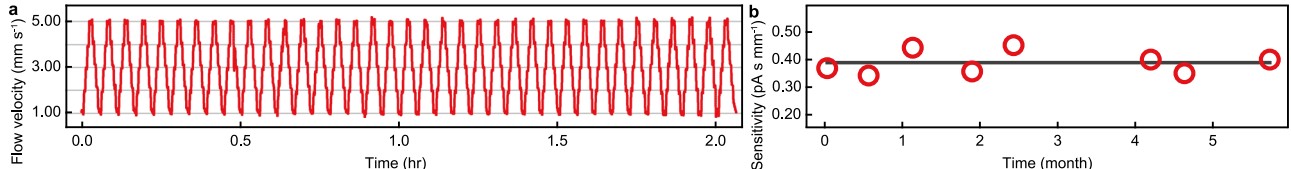

**Fig. 3 Repeatability and stability of the graphene device. a** The measured flow velocity in response to a stepwise flow waveform switching between 1, 2, 3, 4, and 5 mm s$^{-1}$, which are represented by the eye-guiding gray solid lines, every 20 s in turn. **b** Long-term (half-year) stability of sensitivity. The sizes of the error bars (the sensitivity standard deviations based on the uncertainties in current determination over 1-Hz bandwidth) are smaller than the size of the data points. The black line is a constant fit to the data.

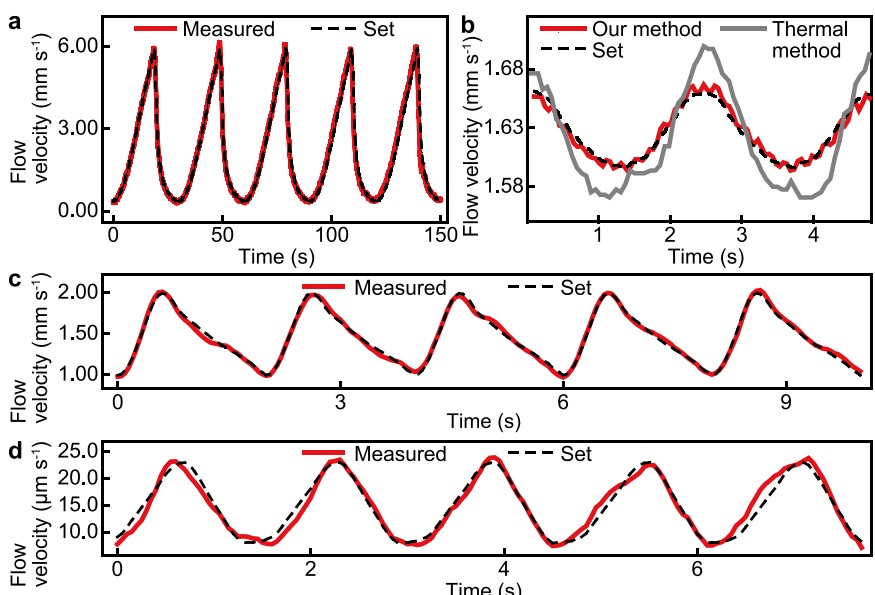

**Fig. 4 Real-time interrogation of pump-driven time-varying whole-blood flows using the graphene devices.** The measured velocities of the sawtooth-like (**a**) and sinuous (**b**) flows and the flows whose waveforms simulate those of the blood flows through murine deep-brain capillaries (**c**) and human retina capillaries (**d**). The velocity of the sinuous flow (**b**) measured by the graphene device shows 6-Hz, <10 μm s$^{-1}$ periodic steps generated by the stepping motion of the stepper motor of the syringe pump. In **a**–**d**, the red lines represent the results of our real-time measurement using the graphene devices and the dashed lines represent the flow velocities set by the syringe pump. The gray line in **b** represents the flow velocity measured by using a calorimetric flowmeter.

impacting on the EDL rearrangement at the graphene/PBS interface.

**High-performance real-time flow transduction.** We then investigated the repeatability and long-term stability of the graphene devices and their capability for real-time identification of multiscale amplitude-temporal flow characteristics. We developed a program for automatically picking up charge-transfer signal from a graphene device and providing the corresponding flow-velocity readout. The program (Supplementary Movie 1) functions in real time to communicate with the coulometer to acquire charge-transfer data, extract electrical current by taking numerical time derivative of the transferred charge, smooth the current using a bandwidth-controllable Savitzky–Golay filter, and translate the smoothed current to flow velocity via linear interpolation of the current–flow data of the graphene device being used.

Figure 3a shows the real-time flow velocity measured by a graphene device in response to a continuous five-step blood-flow that lasted for more than 2 h. The measured velocity demonstrates high repeatability with minimal fluctuations (±0.07 mm s$^{-1}$). We also used a graphene device to perform intermittent measurements for periods of 6 months. The blood-flow sensitivity of the device fluctuated around an average value (0.39 pA s mm$^{-1}$) with a standard deviation of ±0.02 pA s mm$^{-1}$, ±5.1% of the average value, as shown in Fig. 3b. The repeatability, stability, and

anti-fouling capability of the graphene devices reflected by the experiments are high, despite that graphene is only a monolayer layer of carbon atoms. Intrinsically, graphene is of atomic thickness and flatness, making it suitable to be fabricated into the planar device configuration easily to be integrated into the microfluidic device with minimal risk of being fouled or clogging the microfluidic channel. Graphene is formed by C–C sp$^2$-hybridized σ bonds with π-band electrons oriented out of the basal plane, which gives rise to high physicochemical stability and high electrochemical inertness in the solutions with physiological conditions (pH 7.0, ionic strength 150 mM, room temperature, etc.).

We used the graphene devices to register rapid, multiscale variations in four pulsatile waveforms of blood flow driven by the syringe pump, the sawtooth-like (0.33–6.00 mm s$^{-1}$; Fig. 4a), the sinuous (1.60–1.66 mm s$^{-1}$; Fig. 4b), the human-retinal-capillary (1.0–2.0 mm s$^{-1}$; Fig. 4c)[26], and the murine-brain-capillary (9.0–23.0 μm s$^{-1}$; Fig. 4d)[27], in addition to a pulsatile waveform of PBS flow (Supplementary Fig. 8). For all the waveforms, the real-time readout of our program well follows the set flow velocity with minimal delay. In comparison, the outcome of a state-of-the-art high-sensitivity calorimetric flowmeter can deviate substantially from the set flow velocity (Fig. 4b), despite that calorimetric flowmeters have been broadly used in low-flow microfluidic research and applications. For the waveform that

simulates the blood in murine deep-brain capillaries (Fig. 4d), the standard deviation of the readout with respect to the set flow velocity is $1.2 \, \mu m \, s^{-1}$, representing a notable performance metric of the graphene devices in identifying small-scale flow features over other electrical flow-sensing technologies. This flow resolution is comparable with high-performance imaging modalities such as real-time fluorescent cerebrovascular imaging $(5–20 \, \mu m \, s^{-1})$[27] and high-speed adaptive optics imaging $(18–70 \, \mu m \, s^{-1})$[26,28], while the graphene devices retain further advantages in the long-term stability, miniaturization, integrability, and cost efficiency.

## Discussion

Our research provides a self-powered strategy for high-performance biofluid-flow interrogation enabled by graphene. The findings pave the way to future researches on all-electronic in vivo flow monitoring in investigating ultra-low-flow $(<10 \, \mu m \, s^{-1})$ life phenomena that are yet to be studied in metabolomics, retinal hemorheology, and neuroscience. The charges transferred into the graphene devices in the measurement are stored in a feedback capacitor, offering a pathway to highly self-sustained systems that simultaneously probe blood flow and harvest energy from the flow. Despite that a single graphene single-microelectrode device is only suitable for quantifying the velocity of flow at a specific position, arrays of the devices can be developed for mapping flow velocity at high spatiotemporal resolution. In using this graphene-enabled flow-sensory modality for measuring the flow of real-world flow samples, noise may be induced in the charge-transfer current signal due to the complexity of the samples (e.g., the presence of bubbles that can generate moving EDL boundaries at graphene interface) and signal-smoothing methods such as Savitzky–Golay filtering and percentile filtering can be used for the noise reduction. Potential fields of applications of our technology include in vivo biofluid mechanics, vascular tissue engineering, and disease-progression surveillance, to name a few[1,2,29–31], particularly when established imaging modalities are less efficient or difficult to be implemented.

## Methods

**Graphene preparation and transferring**. We prepared monolayer graphene via CVD on copper catalytic substrate. A piece of copper foil (99.8% purity; Alfa Aesar) was loaded into a quartz tube (22 mm in I.D.; 4 feet in length) and annealed for 30 min at 1060 °C in the atmosphere of mixed ultrahigh purity (UHP; 99.999%) hydrogen (flow rate = 200 sccm) and UHP (99.999%) argon (flow rate = 500 sccm) for removing oxide residues. Then ambient-pressure chemical vapor deposition was implemented for the growth of monolayer graphene in the mixed gas of UHP hydrogen (flow rate = 3 sccm), UHP argon (flow rate = 500 sccm, and precursor UHP (99.99%) methane (flow rate = 0.5 sccm) at 1035 °C (growth time 20 min).

A piece of the CVD graphene (100–1000 μm in width) was transferred from the copper substrate onto a 0.5-mm-thick acrylic sheet using the low-contamination bubbling method. At first, a 400-nm-thick layer of polymethyl methacrylate (MICROCHEM, 950 PMMA A4) was spun-coated on the CVD graphene on copper substrate at a speed of 1000 rpm. Then the PMMA/graphene/copper film was connected to the cathode of a power supply and immersed in 1 M NaOH solution. An electric current of ~1 A was applied through the film to generate hydrogen bubbles between the graphene and copper and float off the PMMA/graphene film from the copper. The PMMA/graphene film was then transferred onto a 0.5-mm-thick acrylic sheet with the PMMA in contact with the acrylic sheet and the graphene facing out, forming a graphene/PMMA/acrylic structure.

**Device fabrication**. A GCC LaserPro Spirit GLS laser engraver was used to fabricate $500 \times 500 \, \mu m$ micro-channels in an acrylic sheet (500 μm in thickness). Then the microchannel-structured acrylic sheet and a 0.5-mm-thick substrate acrylic sheet were thermally bonded with the assistance of a solution of 80 vol% dichloromethane (DCM, Honeywell Burdick & Jackson 300-4) and 20 vol% 2-propanol (IPA, Fisher Chemical A416-4) at 70 °C for 10 min in a forced convection oven, forming a structured microfluidic module.

We used e-beam evaporation to deposit a 60-nm-thick Cr/Au lead through which the transferred graphene on the graphene/PMMA/acrylic structure can be electrically connected with a coulombmeter. Then the microfluidic module and the graphene/PMMA/acrylic structure were thermally bonded with the assistance of

the DCM/IPA solution for 90 s at about 100 °C using a heat gun. In this process, the PMMA film sandwiched between the graphene and the acrylic module was bonded to the acrylic module. The device was then cleaned by DI water in an ultrasonic bath for 10 min for residual removal.

**Measurement**. In measurement, the graphene single microelectrode in the microfluidic chip was exposed to $1 \times PBS$ (Fisher BP661-50; pH = 7.0, ionic strength = 150 mM) or bovine whole blood (BIOIVT BOV7411) with flow velocity controlled by a picoliter/min syringe pump (Harvard Apparatus 70-3009). The bovine blood was stored at −86 °C and thawed in a water bath to room temperature (25.0 °C) before the measurement. Only the blood samples undergoing one freeze–thaw cycle were used. The PBS solution was prepared before the experiment. Since the graphene charge-transfer current is associated with the hydrolysable groups at graphene defects and the pH of the liquid through the Langmuir–Freundlich isotherm[32], the pH for the blood/PBS was precisely measured and well-controlled in our experiment.

A Keithley 6517b electrometer in the coulombmeter mode (1-fC charge-measurement resolution, settling time <0.1 μs) was used to quantify the charge transfer of the graphene single microelectrode. Prior to the measurement, the noninverting input of the coulombmeter was grounded while the inverting input was connected to a dissipative resistor. To start the measurement, the inverting input was disconnected from the dissipative resistor and connected to the Au/Cr lead, which linked the graphene single microelectrode to a virtual ground so that the charges that transferred into the graphene were stored in a feedback capacitor of the operational amplifier in the coulombmeter and quantified.

We developed a program to acquire the flow-sensory charge signal from the coulometer through an IEEE-488 interface. The program performs time derivative of the measured charge to obtain the charge-transfer current and uses a bandwidth-variable Savitzky–Golay filter based on the least-squares polynomial method to smooth the current in real time. The measured electric current is translated to flow velocities in the program in real time by interpolating the current–flow data set of the device being tested. The flow velocity measured by our graphene devices was in real time compared to that measured by a state-of-the-art high-sensitivity calorimetric flowmeter (Sensirion SLI1000).

Measurements in our experiment were performed at well-controlled ambient temperature (25.0 ± 0.2 °C). The flow to be interrogated was driven through the microfluidic channel for more than 10 min in advance and no entrapment of bubbles was observed at the graphene device during the measurement. After each measurement, PBS was infused by the syringe pump through the device at $5 \, mm \, s^{-1}$ for 1 min. Then DI water was infused through the device at $5 \, cm \, s^{-1}$ for 4 min, followed by infusing air with a speed of $10 \, cm \, s^{-1}$ for 4 min for drying the device. The device was stored in an airtight, opaque container when not in use. We examined the device using optical microscopy each time after/before the measurement: No visible accumulation of residues (e.g., blood clots) was observed on the graphene microelectrodes.

## Data availability

The authors declare that the data supporting the findings of this study are available within the paper and its supplementary information files. Additional data that support the findings of this study are available from the corresponding author upon request.

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

## Acknowledgements

The authors acknowledge J. Nicholson and J. Yin for e-beam evaporation and reactive ion etching in UMass Amherst Silvio O. Conte National Center. J.P. acknowledges support from Department of Defense (DoD), Air Force Office of Scientific Research (AFOSR) (FA9550-20-1-0125) and DoD, Congressionally Directed Medical Research Programs (CDMRP) (W81XWH-19-1-0006).

## Author contributions

J.P. and X.Z. conceived and designed the project. E.C. and X.Z. fabricated the devices. X.F. prepared graphene samples. X.Z. conducted the measurement. J.P. and X.Z. analyzed the experimental results and wrote the manuscript. All authors discussed the results and commented on the manuscript.

## Competing interests

The authors declare no competing interests.
