## [Peer Review File · Nature Communications]

Reviewers' Comments:

Reviewer #1:

Remarks to the Author:

Zhang et.al. reported that the current arising from Faradic charge transfer between graphene and blood flow could be utilized to construct biofluid flow sensor with high sensitivity and stability. The current is linear proportional to the flow velocity and could be modulated by the graphene-liquid interfacial potential, facilitating its practical applications. Overall, this work is interesting and the manuscript is well organized with in-depth analysis. But several issues should be addressed before the consideration of its publication on Nature Communications.

1. Authors contribute the current to the contact electrification and even took it in the title. However, contact electrification is a general and quite complex process, involving not only electron tunneling but also ion exchanges and other effects. So it could be better to take a more precise description.
2. In the supplementary, Ref.S9, S10 are classified into the triboelectricity group, which is definitely wrong. The mechanism of the work presented in Ref.S9, S10 is the moving boundary of EDL. Please refer to Nature Nanotechnology, 13, 1109-1119, 2018 for proper summary and classification of works in this area.
3. It's not clearly explained that why there is a constant current flow through the graphene/liquid interface during the measurements, i.e., why the system cannot reach its equilibrium state. Is it because the presence of a capacitor in the measurement circuit, and it take a significant long time to charge the capacitor? If so, how long the sensor could work properly without discharging the capacitor?
4. It's claimed that the tunneling current is highly dominated by the defect states in graphene. Then, treatments, such as moderate plasma etch, could increase the defect density and improve the sensor's performance. If so, it would be a strong support to the authors' explanation.
5. At the end of the second paragraph in the Results and Discussion section, several data were directly cited from their previous work, where Si/SiO₂ and phosphate buffer solution was used as substrate and liquid, respectively. However, graphene's properties are quite sensitive to the substrates/doping level. So, it's not reasonable to adapt these data directly. Moreover, I think it's a mistake when mention "conductance of the CVD graphene", which was supposed be "conductance of the CVD graphene/liquid interface".
6. How to control the surface charge density of the glass electrode? If it's not controllable, then it would hinder its practical applications. Also, how it's connected to the graphene?
7. The resolution of the flow sensor is highly dependent on the bandwidth, but the way to modulate the measurement bandwidth is not given.

Reviewer #2:

Remarks to the Author:

Zhang et al's manuscript, entitled "Flow-Sensory Contact Electrification of Graphene", presents a graphene based electrical flow sensor for monitoring flow rates in microfluidic devices and various other medical apparatus used for hemodynamic analysis. The flow sensor ultrathin (atomic layers) and demonstrated to be stable over 6-months. The comprehensive testing under different flow conditions, and signal resolution are impressive and the manuscript is well written. I do, however, have a few questions and comments about the graphene device and performance. The authors should address these concerns to make this work suitable for publication in highly selective Nature Communications journal.

Major comments:

1. Figure 1 provides overview of graphene flow sensor, but it is difficult to understand the flow profile (plug flow vs poiseuille?) and the effects fluid boundary layer thickness on flow estimation. It would be helpful to include schematic drawing detailing the different flow profiles.
2. It would be useful to include an image in Figure 1 of the graphene sensor and experimental

setup (as in Figure S1) with magnified view of the graphene sensor, electrode contacts, and microfluidic channel. Schematic drawing in Figure 1 needs more detail to understand the experimental setup.

3. The authors should include more discussion about the cleaning procedures over months. It is impressive to see the devices operates over 6 months. How does the graphene maintain its performance without fouling caused by blood clots, cell growths, etc.

4. The authors should also elaborate on effects of pH, temperature viscosity on their performance curves. The results are impressive. Any additional insight about changes in performance would be useful to incorporate.

5. What are the limitations of this system relative to other thermal, optical based systems. The authors do a good job of presenting the prior work in flow sensing (Table S1). They should include brief discussion of the limitations of their graphene flow system in the Discussion section.

Reviewer #3:

None

Reviewer #4:

None

Reviewer comments are shown in *italics* and our responses are in plain text.

Responses to Reviewer #1:

Zhang et.al. reported that the current arising from Faradic charge transfer between graphene and blood flow could be utilized to construct biofluid flow sensor with high sensitivity and stability. The current is linear proportional to the flow velocity and could be modulated by the graphene-liquid interfacial potential, facilitating its practical applications. Overall, this work is interesting and the manuscript is well organized with in-depth analysis. But several issues should be addressed before the consideration of its publication on Nature Communications.

We are grateful to the reviewer for the thorough review and helping to improve the paper. We are pleased that the reviewer thought that of our work is interesting and the manuscript is well organized with in-depth analysis.

Q1. Authors contribute the current to the contact electrification and even took it in the title. However, contact electrification is a general and quite complex process, involving not only electron tunneling but also ion exchanges and other effects. So it could be better to take a more precise description.

According to two highly cited papers (McCarty L. S. & Whitesides G. M. *Angewandte Chemie* **47**, 2188-2207 (2008), Ref. 14 in the manuscript; Lin *et al.* *Angewandte Chemie* **52**, 12545-

12549 (2013)), contact electrification, or tribocharging, is defined by the charge transferring from one material to another. In our experiment, we used a feedback capacitor of an operational amplifier to collect charges transferring between graphene and blood/PBS flow and demonstrated the relationship between the charge-transfer current with the flow velocity of the blood/PBS. The generation of this flow sensory charge-transfer current right fits the definition of contact electrification. We also noticed that many papers used “contact electrification”, “flow electrification”, or “triboelectric” in their papers/titles to describe the current generated when materials were in contact with each other, regardless of the various mechanisms for generating the current: *e.g.*, Washabaugh A. P. & Zahn M. *IEEE Transactions on Dielectrics and Electrical Insulation* **3**, 161-181 (1996), McCarty L. S. & Whitesides G. M. *Angewandte Chemie* **47**, 2188-2207 (2008) (Ref. 14 in the manuscript), Lin Z.-H. *et al.* *Angewandte Chemie* **52**, 12545-12549 (2013), Lin Z.-H. *et al.* *Advanced Materials* **26**, 4690-4696 (2014), Chen J. *et al.* *ACS Nano* **10**, 8104-8112 (2016), Liu J. *et al.* *Nature Nanotechnology* **13**, 112-116 (2018), Wang Z. L. & Wang A. C. *Materials Today* **30**, 34-51 (2019), Nauruzbayeva J. *et al.* *Nature Communications* **11**, 5285 (2020) *etc.*

As discussed in the manuscript, our experimental results and model-based analysis well agree with the quantum-tunneling picture of the charge-transfer current at graphene-water interfaces. In our previous published paper (Ping, J. & Johnson, A. T. C. *Applied Physics Letter* **109**, 013103 (2016), Ref. 18 in the manuscript), the charge-transfer current is associated to charge-transfer processes through graphene defect states. The defect density determined by the Raman spectroscopy ($1.44 \times 10^{17} \text{ cm}^{-2}$) is in excellent agreement with the defect density value ($\sim 10^{17} \text{ cm}^{-2}$) evaluated by the specific charge-transfer conductance of the CVD graphene ($1.5 \times 10^{-9} \Omega^{-1}$

mm⁻²), provided that the electrical conductance corresponding to a typical electron tunneling process is $\sim 10^{-24} \Omega^{-1}$ (McCarty, L. S. & Whitesides, G. M. *Angewandte Chemie, International Edition* **47**, 2188-2207 (2007), Ref. 14 in the manuscript). In the revised paper (Results and Discussions, second paragraph), we are happy to highlight this point.

The reviewer mentioned in Q4 that associating the increase in the charge-transfer current of a graphene single microelectrode to the enhancement of disorder density of the graphene by plasma etching can ideally be used to demonstrate the tunneling origin of the charge-transfer current at graphene defects. In our response to Q4, we measured the charge-transfer current of a graphene single microelectrode that was treated by oxygen plasma for different periods. The result clearly indicates that the charge-transfer current depends positively on the period of the plasma treatment. We are grateful to Reviewer #1 for making the constructive and inspiring suggestion.

Q2. In the supplementary, Ref.S9, S10 are classified into the triboelectricity group, which is definitely wrong. The mechanism of the work presented in Ref.S9, S10 is the moving boundary of EDL. Please refer to Nature Nanotechnology, 13, 1109-1119, 2018 for proper summary and classification of works in this area.

We are happy to make changes to the terminology as per the suggestions made by the reviewer. We have made corresponding clarification in the revised manuscript (Results and Discussion, fourth paragraph) and in the revised supplementary information (Supplementary Table 1). We

have also cited the *Nature Nanotechnology* paper in the revised manuscript (Ref. 21 in Results and Discussion, fourth paragraph).

Q3. It's not clearly explained that why there is a constant current flow through the graphene/liquid interface during the measurements, i.e., why the system cannot reach its equilibrium state. Is it because the presence of a capacitor in the measurement circuit, and it take a significant long time to charge the capacitor? If so, how long the sensor could work properly without discharging the capacitor?

The capacitor in the circuit is a feedback capacitor of the operational amplifier and its sole function is to collect the charges that transfer into graphene and to generate output voltage that is indicative to the amount of the collected charges. An analog to digital converter (ADC) measures the output voltage (U) of the feedback capacitor (C) and transfers the voltage to the quantity of charge ($Q = C \times U$). The measurement can be taken continuously until the voltage of the capacitor exceeds the measurement range of the ADC. In our experiment, we used a 20-nC full measurement range, which could work continuously from 1.4 hours (Fig. 2c in the manuscript, G5 current) to 120 hours (Fig. 2c in the manuscript, inert current), depending on the current level. (The feedback capacitor can be controllably discharged to start a new measurement cycle.) The non-inverting input of the operational amplifier was grounded, thus the graphene was virtually grounded and held at a constant electrostatic potential. As such, the charge that were collected by the feedback capacitor did not generate impact on the charge transfer at the graphene/water interfaces. Rather, the charge-transfer process is determined by the graphene defect density and

can be modulated by the chemical and electrostatic condition in the environment of the virtually grounded graphene in the liquid.

There is a constant current flow between the graphene single microelectrode and the liquid because the pH of the liquid is held at a constant value. In our previously published work (Ping J et al. *Small* **13**, 1700564 (2017)), we demonstrated the charge-transfer current at graphene/water interface is associated to the reversible hydrolysis of the chemical groups of the graphene defects and depends on the isoelectric point of the chemical groups. Therefore, a constant pH of the liquid leads to a steady charging status of the chemical groups and therefore generates constant charge-transfer current at the graphene/water interface. Also, the graphene and the liquid together form an open system in thermodynamic and electrostatic equilibrium with the external environment, allowing continuous and sustained charge collection by the graphene at constant pH of the liquid. After the equilibrium is built, for a typical binding energy of electron (*e.g.*, 5 eV) at a graphene defect state, the probability for an individual electron tunnelling across a 2-nm gap is only about 10^{-20} (Harper, W. R. *Contact and Frictional Electrification*, Laplacian Press, 1998; McCarty, L. S. & Whitesides, G. M. *Angewandte Chemie, International Edition* **47**, 2188-2207 (2007), Ref. 14 in the manuscript). Charge transferred at such low level can be readily compensated by the charge provided by the environment without causing variation of the pH of the liquid. As such, the state of the graphene/liquid interface can always be considered to be stable and the EDL is always well-defined at the ideal non-faradiac graphene/water interfaces in our measurement. We are happy to highlight in the Methods–Coulometric Measurement section of the revised manuscript that the pH of the blood and PBS were precisely determined and well controlled.

Q4. *It's claimed that the tunneling current is highly dominated by the defect states in graphene. Then, treatments, such as moderate plasma etch, could increase the defect density and improve the sensor's performance. If so, it would be a strong support to the authors' explanation.*

According to the scientific understanding of the charge-transfer current at graphene/water interface, increasing the period of exposing graphene to plasma can increase the defect density of the graphene and hence the charge-transfer current level. (In the manuscript, we have demonstrated that the charge transfer is associated to the defect states instead of the edge states at graphene basal plane in *Supplementary Fig. 1*.) In this sense, as per the reviewer's suggestion, we measured the charge-transfer current of a graphene single microelectrode that was treated by oxygen plasma (10-W power, 50-sccm oxygen flow rate, 4-mtorr pressure) for different periods. As shown by the data below this paragraph, the result clearly indicates that the scaled current (the charge-transfer current of a graphene microelectrode normalized by that of the graphene prior to oxygen plasma treatment) is enhanced by up to 3× as the disorder density increases, before the graphene was destroyed by the plasma and the scaled current reached to zero. We thank the reviewer for making this suggestion and have added these data to *Supplementary Fig. 2* in the *Supplementary Information* of the revised manuscript.

Although the charge-transfer current of the graphene single microelectrode is straightforwardly associated to the defect density of graphene that is determined by the period of plasma treatment, the performance of the graphene flow sensors, such as the sensitivity and stability, relies on many other parameters besides the graphene defect density. Those parameters are variable by the plasma treatment: the types of the hydrolysable chemical groups at graphene defects created by the plasma etching and the respective proportions of the groups, the stability and steadiness of the plasma-created defect states in water flow, the noise level of the current that depends on the created chemical groups, *etc.* As indicated in published papers (*e.g.*, Merenda A. *et al. Scientific Reports* **6**, 31565 (2016)), carbon-based nanomaterials treated by oxygen plasma are of high complexity in the type of the created chemical groups and their respective proportions, as well as in their temporal stability. Engineering the defects of graphene and hence its charge-transfer efficiency at high precision, stability, and uniformity remains an open question out of the scope of this paper (Kaplan A. *et al. Chemical Society Reviews* **46**, 4530-4571 (2017)).

In the manuscript, we discussed comprehensive experiments we have performed to build scientific understanding of the flow sensory properties of the graphene single microelectrodes and to the performance of the graphene single microelectrodes for flow sensing. We have

measured the performance curves of different types of liquid (blood and PBS; Fig. 2a in the main text and *Supplementary Fig. 5a* for blood and *Supplementary Fig. 7b* for PBS). We have changed the electrostatic potential at graphene by using a glass electrode with varied surface charge density (Fig. 2c and d in the main text). We have investigated the effect of the fluid gauge pressure and the graphene edge defect density on the charge-transfer current (*Supplementary Fig. 1 and 4*). We also have varied the bandwidth in measurement and built a bandwidth–current relationship (Fig. 2e). The results of these experiments are in good agreement with the scenario we discussed in the manuscript that flow sensory properties of our graphene flow sensors is a result of EDL rearrangement. Using the glass electrode also significantly enhanced the sensitivity of the graphene flow sensors to $0.49 \pm 0.01 \mu\text{m s}^{-1}$ (Fig. 2d and *Supplementary Fig. 6*) that is a two-orders-of-magnitude improvement compared with previous device-based flow sensors.

Q5. At the end of the second paragraph in the Results and Discussion section, several data were directly cited from their previous work, where Si/SiO₂ and phosphate buffer solution was used as substrate and liquid, respectively. However, graphene's properties are quite sensitive to the substrates/doping level. So, it's not reasonable to adapt these data directly. Moreover, I think it's a mistake when mention "conductance of the CVD graphene", which was supposed be "conductance of the CVD graphene/liquid interface".

The discussion in the second paragraph of the Results and Discussion section was only for providing background information on the tunneling origin of the charge-transfer current and was

not linked to or used to interpret the data of this work. (The discussion is based on previously published results: one from a highly cited paper on contact electrification (McCarty, L. S. & Whitesides, G. M. *Angewandte Chemie, International Edition* **47**, 2188-2207 (2007), Ref. 14 in the manuscript) and one from our own group (Ping, J. & Johnson, A. T. C. *Applied Physics Letter* **109**, 013103 (2016), Ref. 18 in the manuscript).) Also, the variable factors mentioned by the reviewer do not challenge the tunneling origin of the charge-transfer current.

We agree with the reviewer in the sense that the mobility and carrier density of graphene, together with some other properties (*e.g.*, electrostatic and electron transport properties), are sensitive to the Fermi level of the graphene and thus depend on the substrate and the doping level of graphene. But in this manuscript, the graphene single microelectrode is virtually grounded through the operational amplifier, as mentioned in our response to Q3 of the reviewer, which leads to a constant Fermi level of the graphene. The measured current is dominated by the heterogeneous charge-transfer process through the defects of graphene at graphene-liquid interface, rather than by the in-plane electron transport process. In fact, Faradaic charge transfer occurred through the series combination of the solution diffuse layer resistance ($\sim 10\text{ k}\Omega$), the graphene-solution interface charge-transfer resistance ($R_{ct} \sim 100\text{ G}\Omega$), the sheet resistance of graphene ($\sim 10^2 - 10^3\ \Omega/\square$), and the graphene-gold contact resistance ($\sim 10\ \Omega$). Therefore, the charge transfer rate is dominated by the charge transfer resistance R_{ct} and, in consequence, the defect density of graphene which does not depend on the substrate.

In the same way, the graphene defect density is independent to the liquid whose flow is measured. In our experiment, we used a same graphene microelectrode to quantify the flow

velocity of both PBS and bovine blood and the result demonstrates that the trends of the performance curve for PBS and bovine blood are comparable.

Although in the second paragraph of the Results and Discussion section, the results of our previously published paper is only used for providing background information on the tunneling origin of the charge-transfer current and not for interpreting the data of this work, we were inspired by the comment of the reviewer and noticed that the level of the charge-transfer current measured in this work is in line with that in the published paper. The process and setup for preparing graphene through CVD in the experiment of this manuscript are similar to our previously published work (Ping, J. & Johnson, A. T. C. *Applied Physics Letter* **109**, 013103 (2016), Ref. 18 in the manuscript); the graphene samples in both works were not treated or doped. In the previous work, the specific charge-transfer conductance of the CVD graphene on the SiO₂/Si substrate that was of steady charging state (electrostatic potential) in water and modulated the graphene charge-transfer current is σ_{ct} ($1.5 \times 10^{-9} \Omega^{-1} \text{mm}^{-2}$) at zero flow velocity. In this manuscript, the zero-velocity current for the system with glass electrode is at the level of \sim pA, leading to σ_{ct} of $\sim 10^{-9} \Omega^{-1} \text{mm}^{-2}$, well agreeing with that ($1.5 \times 10^{-9} \Omega^{-1} \text{mm}^{-2}$) in our previously published work. We are happy to highlight this point in the revised manuscript (Results and Discussions, fifth paragraph).

We thank the reviewer for pointing out the omission of using the terminology of “specific conductance”. In the revised manuscript, we have used “specific charge-transfer conductance” to match the “charge-transfer resistance” that is broadly used.

Q6. How to control the surface charge density of the glass electrode? If it's not controllable, then it would hinder its practical applications. Also, how it's connected to the graphene?

There are many different ways that can be used to control the surface charge density of the glass electrode, for example, through surface modification, ionization of the surface, and varying the condition (e.g. pH) of the liquid (Geisler T. *et al. Journal of Non-Crystalline Solids* **356**, 1458-1465 (2010), Dultz S. *et al. Chemical Geology* **426**, 71-48 (2016), Gu Y. & Li D. *Journal of Colloid and Interface Science* **226**, 328-339 (2000)). In our experiment (Fig. 2c), the surface charge density of the glass electrode was controlled by surface ionization. In the revised manuscript (Results and Discussion, fifth paragraph), we are happy to highlight the surface ionization mechanism for controlling the surface-charge density of the glass electrode. We thank the reviewer for the comment.

The glass was not in contact with the graphene directly. Rather, it functioned as a counter electrode with steady charging status that modulated the electrostatic potential at the graphene/water interface. The charging properties of glass electrodes for modulating electrostatic potential can be found in many papers that are cited in the manuscript: Jednacak-Bisčan J. & Pravdić V. *Journal of Colloid and Interface Science* **90**, 44-50 (1982), Elimelech M., Jia X., Gregory J., & Williams R. (1998) *Particle Deposition and Aggregation: Measurement, Modelling and Simulation* (Elsevier), and Al Mahrouqi D., Vinogradov J., & Jackson M. D. *Advances in Colloid and Interface Science* **240**, 60-76 (2017) (Ref. 23–25 in the manuscript, respectively). Our results of the polarity and magnitude of the charge-transfer current of the

graphene single-microelectrode flow sensor as a function of flow velocity are well in line with the understanding based on the modulating effect arising from the variation of the surface charge of the glass electrode, as described in the inset energy profiles in Fig. 2c and in the discussion in paragraph 5–7 of the Results and Discussion Section.

Q7. The resolution of the flow sensor is highly dependent on the bandwidth, but the way to modulate the measurement bandwidth is not given.

We agree with the reviewer that the resolution of the flow sensor is highly dependent on the bandwidth. As described in the Methods–Coulometric Measurement section and the ninth paragraph of the Results and Discussion section of the manuscript, a real-time Savitzky-Golay filter was used in our experiment to control the bandwidth of the data and smooth the data, and to control the bandwidth to a specific value for achieving a corresponding resolution. The resolution–bandwidth relationship is shown in Fig. 2e of the manuscript. We also showed the real-time smoothing with controlled bandwidth in the *Supplementary Video 1*. At the beginning of the Results and Discussion (second paragraph) in the revised manuscript, we are happy to highlight that we used the Savitzky-Golay method for bandwidth controlling.

Responses to Reviewer #2:

Zhang et al's manuscript, entitled "Flow-Sensory Contact Electrification of Graphene", presents a graphene based electrical flow sensor for monitoring flow rates in microfluidic devices and various other medical apparatus used for hemodynamic analysis. The flow sensor ultrathin (atomic layers) and demonstrated to be stable over 6-months. The comprehensive testing under different flow conditions, and signal resolution are impressive and the manuscript is well written. I do, however, have a few questions and comments about the graphene device and performance. The authors should address these concerns to make this work suitable for publication in highly selective Nature Communications journal.

We are grateful to Reviewer #2 for the thorough review and helping to improve the paper. We are pleased that Reviewer #2 thought that of our results are impressive and the manuscript is well written. Reviewer #2 also mentioned one of the advantages of our graphene flow sensors that they are ultrathin and of atomic thickness. We are happy to highlight this in the Introduction section.

Q1. Figure 1 provides overview of graphene flow sensor, but it is difficult to understand the flow profile (plug flow vs poiseuille?) and the effects fluid boundary layer thickness on flow estimation. It would be helpful to include schematic drawing detailing the different flow profiles.

We thank the reviewer for the suggestion. Near the wall of a microfluidic channel at graphene EDL, pump-driven blood/PBS can be considered as poiseuille flow (Bagchi P. *Biophysical Journal* **92**, 1858-1877 (2007)). Since the charge-transfer is governed by the electrostatic potential at the graphene EDL, which is modulated by the flow, the graphene single-microelectrode flow sensors are expected to show comparable current–flow relationships for blood and PBS, only with difference in the magnitude of sensitivity (current to flow ratio) arising from the difference in the viscosities between the blood and PBS. This is consistent with our experimental outcome. We also noticed that investigations and a quantitative understanding of the condition of blood/PBS flow on graphene, including the position of the shear plane and the scale of the slip length, remains an open question (Celebi, A. T. *et al. Microfluidics and Nanofluidics* **22**, 7 (2018), Xie, Q. *et al. Nature Nanotechnology* **13**, 238-245 (2018)). As such, in Fig. 1a in the revised manuscript, we have added a conceptual representation of the profiles of the flow and the electrostatic potential above graphene, in light of previous research (Kirby, B. *Electrophoresis* **25**, 187-202 (2004)).

Q2. *It would be useful to include an image in Figure 1 of the graphene sensor and experimental setup (as in Figure S1) with magnified view of the graphene sensor, electrode contacts, and microfluidic channel. Schematic drawing in Figure 1 needs more detail to understand the experimental setup.*

We thank the reviewer for the suggestion. In the revised version of the manuscript, we updated Fig. 1a accordingly. We have added the image of the whole device (*Supplementary Fig. 1* was

removed in the updated version of the Supplementary Information), indicated the measurement setup based on the device, and built a magnified view of the graphene electrode on the device. As mentioned in our response to Q1 of the reviewer, we have also added a conceptual representation of the profiles of the flow and the electrostatic potential in the liquid above graphene in Fig. 1a in the revised manuscript. Figure 1a is more informative and we are grateful to the reviewer for making the constructive and inspiring suggestion.

Q3. The authors should include more discussion about the cleaning procedures over months. It is impressive to see the devices operates over 6 months. How does the graphene maintain its performance without fouling caused by blood clots, cell growths, etc.

We thank the reviewer for the remarks. We are also very excited about the immunity of the performance of the graphene microelectrodes to biofouling demonstrated in our experiment and we are optimistic about the potential application of this property of graphene in further applications.

As describe in our manuscript (Results and Discussion, paragraph 10), we used a graphene device to perform intermittent measurements for periods of six months. After each measurement for obtaining the sensitivity of the device (Fig. 3b), clean PBS was infused by the syringe pump through the device at 5 mm s^{-1} for 1 min. Then DI water was infused through the device at 5 cm s^{-1} for 4 min, followed by infusing air with a speed of 10 cm s^{-1} for 4 min for drying the device. The device was stored in an airtight, opaque container when not in use. We examined the device

using optical microscopy each time after/before the measurement: no visible accumulation of residues (*e.g.*, blood clots) was observed on the graphene microelectrodes. We are happy to add this information to the last paragraph in the Methods–Coulometric Measurement section of the revised manuscript.

As in our discussion in the manuscript (paragraph 10 of the Results and Discussion section), the immunity of the performance of the graphene single-microelectrode devices to biofouling is associated to the intrinsic flatness and single-atomic thickness of graphene, which delivers low possibility for the accumulation of bio-contamination on the devices. Since the contamination is not accumulatable, rapid absorption equilibrium can be built after graphene was in contact with biosamples (like serum and blood). Indeed, as demonstrated in our previously published result (Ping J et al. *Small* **13**, 1700564 (2017)), an absorption equilibrium can be built in seconds and led to steady charge-transfer current.

Q4. The authors should also elaborate on effects of pH, temperature viscosity on their performance curves. The results are impressive. Any additional insight about changes in performance would be useful to incorporate.

We thank the reviewer for the remarks. The manuscript includes extensive experiments we have performed to build scientific understanding of the flow sensory properties of the graphene single microelectrodes and to demonstrate the outstanding performance of the graphene devices for flow sensing, as in our response to Q4 of Reviwer #1. We have measured different types of

liquid with different viscosity (blood and PBS; Fig. 2a in the main text and *Supplementary Fig. 5a* for blood and *Supplementary Fig. 7b* for PBS). We have changed the electrostatic potential at graphene by using a glass electrode with varied surface charge density (Fig. 2c and d in the main text). We have investigated the effect of the fluid gauge pressure and the graphene edge defect density on the charge-transfer current (*Supplementary Fig. 1* and 4). We also have built a relationship between the current with bandwidth in measurement (Fig. 2e). The results of these experiments reflect the performance of the graphene flow sensors in the different conditions and are well in line with the EDL-rearrangement origin of the flow sensory properties.

As discussed in the manuscript, the viscosity between blood and PBS induces the difference in the magnitude of the charge-transfer current of the graphene single microelectrodes for measuring the flows of blood and PBS, and hence the difference in the magnitude of the sensitivity in the performance curves (Fig. 2a and *Supplementary Fig. 5a* in the main text for blood and *Supplementary Fig. 7b* for PBS). A quantitative understanding on the impact of the viscosity of the flow sensory properties of the graphene single microelectrode relies highly on the condition of the flow on graphene, including the position of the shear plane and the scale of the slip length and other parameters, which, as we mentioned in our response to Q1 of the reviewer, remains an open question (Celebi, A. T. *et al. Microfluidics and Nanofluidics* **22**, 7 (2018), Xie, Q. *et al. Nature Nanotechnology* **13**, 238-245 (2018)).

The flow sensory properties of the graphene single microelectrode also depend on the electrostatic potential at the graphene/water interface. Varying the surface charge density of a glass electrode effectively changes the performance curves of the graphene flow sensors, which

is shown in Fig. 2a and b and reflected in Fig. 2c. These results are in good agreement with the first-order EDL-rearrangement scenario for the flow-sensory charge-transfer current of the graphene devices.

As in our responses to Q1 and Q3 of Reviewer #1, in our previously published work (Ping J et al. *Small* **13**, 1700564 (2017)), we demonstrated that the charge-transfer current at the interface between PBS and a graphene single microelectrode is associated to the hydrolysis of the chemical groups of the graphene defects and depends on the pH of the PBS and the isoelectric point of the chemical groups through the Langmuir-Freundlich isotherm. In the Methods–Coulometric Measurement section of the revised manuscript, we are happy to mention the association between the graphene charge-transfer current and the hydrolyzable groups and that the pH of the blood and PBS was precisely determined and well controlled. We also cited the *Small* paper in the revised manuscript. Our manuscript focuses on the application of the graphene flow sensors for biofluids (like blood). Biofluids can be view as high-performance buffers with constant pH values and changing the pH of biofluids can generate impact on the samples, such as the lysis of the blood cells, and vary the structure and ingredients of the biofluids which will further change the viscosity of the biofluids and the charge-transfer current at the graphene-water interface (Reinhart W. *et al. Journal of Critical Care* **17**, 68-73 (2002)). The pH variation can also create negative impact on the stability of the acrylic devices (Swift T. *et al. Soft Matter* **12**, 2542-2549 (2016)).

Varying the temperature of the flow in our system can also induce complexity to the measurement and change many parameters in the system: the energy profile for the quantun

tunneling efficiency, the hydrolysis of chemical groups at graphene defects, the charging status and of the acrylic devices (Cui L. *et al. Journal of Electrostatics* **44**, 61-65 (1998)), the chemical equilibrium of the liquid, and the viscosity of the liquid (Çinar Y. *et al. American Journal of Hypertension* **14**, 433-438 (2001)). Therefore, the association of the temperature to the charge-transfer current and the performance curves of the graphene flow sensors is not straightforward and a scientific understanding remains an open question. Measurements in our experiment were performed at well-controlled ambient temperature (25.0 ± 0.2 °C). We are happy to mention about this in the Methods–Coulometric Measurement section of the revised manuscript.

Q5. What are the limitations of this system relative to other thermal, optical based systems. The authors do a good job of presenting the prior work in flow sensing (Table S1). They should include brief discussion of the limitations of their graphene flow system in the Discussion section.

We acknowledge the remarks of the reviewer. In this paper, we discussed a graphene-enabled flow sensory modality that demonstrates key performance metrics orders of magnitude higher than other electrical approaches. We also demonstrated in Fig. 4b in the manuscript the advantage of using of graphene-based flow sensors over thermal methods which have been broadly used in microfluidic researches and applications. Furthermore, the velocity resolution ($\sim \mu\text{m s}^{-1}$) of our devices is comparable, if not superior, to that of state-of-the-art optical systems but our devices retain further advantages in the long-term stability, miniaturization, integrability, and cost and energy efficiency, as discussed in the manuscript.

Like all other single-device electrical flow sensors, the proof-of-principle graphene-single-microelectrode devices described in the manuscript are only suitable for quantifying flow velocity at a specific position in a flow channel while optical methods can be used to map fluidic velocity over a broad scale. Despite that, in future researches, arrays of graphene single microelectrodes can be potentially developed and used for the electrical mapping of flow-velocity field at high spatiotemporal resolutions. We are happy to mention about this in the conclusion paragraph of the revised manuscript.

Reviewers' Comments:

Reviewer #1:

Remarks to the Author:

The authors have addressed my concerns except the first one. In my opinion, "contact electrification" is a quietly wide concept, including charge separation between solid-solid, solid-liquid, solid-gas, etc. Instead, flow electrification, as mentioned by the authors, is more specific and proper for this work.

Reviewer #2:

Remarks to the Author:

The authors have addressed my comments about the figures and biofouling. These modifications have helped improve the manuscript.

I have a follow up comment about the limitations of the device. In light of the electric current densities applied, please address the presence and effects of (micro)bubble formation at the microelectrode interfaces at the operating frequencies.

Reviewer comments are shown in *italics* and our responses are in plain text.

Responses to Reviewer #1:

The authors have addressed my concerns except the first one. In my opinion, “contact electrification” is a quietly wide concept, including charge separation between solid-solid, solid-liquid, solid-gas, etc. Instead, flow electrification, as mentioned by the authors, is more specific and proper for this work.

We are grateful to Reviewer #1 for the thorough review and helping to improve the paper.

Reviewer #1 commented about the potential unclarity that may arise from using “contact electrification” in the manuscript since the term may refer to charge transfer between matters of different phases (solid–solid, solid–liquid, solid–gas, *etc.*). In the revised manuscript (Results and Discussion, fourth paragraph), we have added “graphene-water” to modify “contact-electrification current” where misunderstanding might happen. After the revision, it should be clear that the contact electrification refers to the charge-transfer process at graphene/water interface:

- Results and Discussion, fourth paragraph: “the electron-tunneling origin of the contact-electrification current” has been changed to “the electron-tunneling origin of the **graphene-water** contact-electrification current”.

We have also added “flow” in Caption of Figure 1 for highlighting the condition of the liquid:

- Caption of Figure 1: “Coulometric measurement of contact-electrification charge transfer between whole blood and graphene” has been changed to “Coulometric measurement of contact-electrification charge transfer between **whole-blood flow** and graphene”

We noticed that at other places where the term of “contact electrification” are also used, including the title, “electrification” is well modified or restricted by terms and it should be clear that the “electrification” refers to charge transfer at graphene/liquid interface

(modifications/restrictions are highlighted in **bold** here):

- Title: **flow-sensory** contact electrification of graphene
- Results and Discussion, first paragraph: graphene that harvests charge **from flowing blood** through contact electrification
- Results and Discussion, second paragraph: the electron-transfer mechanism for **solid/water** contact electrification (Note that this paragraph is for providing background information of the charge-transfer current and the flow response of the current is not discussed in this paragraph.)
- Results and Discussion, eighth paragraph: the contact-electrification charge transfer for **the flow of phosphate buffered saline (PBS) at different velocities**

We thank the reviewer for making the suggestion of using the “flow-electrification”. We noticed that some other papers using “flow-electrification” to describe the electricity that is generated when an object is moved over another one on the surface (Washabaugh A. P. & Zahn M. *IEEE Transactions on Dielectrics and Electrical Insulation* **3**, 161-181 (1996)), as mentioned in our response that accompanied to the first version of the revised manuscript. But our research is

slightly different. According to our scientific understanding of the graphene-water contact electrification in the Results and Discussion section of the manuscript, the charge-transfer current is not induced by flow: The electrification occurs and the charge-transfer current is generated when graphene and liquid are in contact with each other (*i.e.*, contact electrification); the charge-transfer current is modulated by the flow and demonstrates linear relationship with the flow velocity.

Responses to Reviewer #2:

The authors have addressed my comments about the figures and biofouling. These modifications have helped improve the manuscript.

We are grateful to Reviewer #2 for the thorough review and helping to improve the paper. We are pleased that Reviewer #2 thought that our modifications have helped improve the manuscript.

I have a follow up comment about the limitations of the device. In light of the electric current densities applied, please address the presence and effects of (micro)bubble formation at the microelectrode interfaces at the operating frequencies.

Reviewer #2 commented about the possibility of (micro)bubble formation at the microelectrode interfaces and its potential impact on the limitations of implementing the graphene devices for sensing flow.

As discussed in the manuscript (Introduction, first paragraph), one of the advantage of our graphene flow sensors is that they are suitable to be used for measuring the flow rate of continuous flow, over typical flow sensors that are enabled by the cyclical formation of the electrical double layer (EDL) of aqueous solution, *i.e.*, the moving EDL boundaries, and are only suitable for liquid/gas dual-phase mixtures. In our experiment, we measured the charge-transfer current for continuous flows in the absence of bubbles: Before each measure, the flow to be interrogated was driven through the microfluidic channel for more than 10 minutes and no entrapment of bubbles was observed at the graphene device during the measurement; compared to high current density at which conventional electrochemical devices or triboelectric devices are operated to generate nucleation of bubbles at the solid-water interface (*e.g.*, $>0.1 \text{ A mm}^{-2}$ in Fernández, D. *et al.* Bubble formation at gas-evolving microelectrode. *Langmuir* **30**, 13065-13074 (2014)), the charge-transfer current density of our graphene devices ($\sim \text{nA mm}^{-2}$) is orders of magnitude lower and we did not observe formation of bubbles at the graphene devices in the measurements. Those are in perfect agreement with the stability of our graphene devices, the low noise in the charge-transfer current signal, and the consistency of the measurement outcomes. We are happy to add this information to the Methods-Measurement section in the revised manuscript. (For clarity, we replaced the title of the subsection from “Coulometric Measurement” to “Measurement” in the revised manuscript.)

We agree with the reviewer that, when used for an unknown complex flow in real-world applications, the response of our graphene flow sensors, like all other electrical flow sensors, can be interfered by potential bubbles in the flow, due to the electrical response of the devices to moving EDL boundaries. The moving EDL boundaries can generate cyclic potential variation at the graphene/water interface, leading to fluctuations in a shot noise-like form in the electrical signal of graphene devices (Li, X. *et al.* Self-powered triboelectric nanosensor for microfluidics and cavity-confined solution chemistry. *ACS Nano* **9**, 11056-11063 (2015); Chen, J. *et al.* Self-powered triboelectric micro liquid/gas flow sensor for microfluidics. *ACS Nano* **10**, 8104-8112 (2016); Yin, J. *et al.* Generating electricity by moving a droplet of ionic liquid along graphene. *Nature Nanotechnology* **9**, 378-383 (2014); Yin, J. *et al.* Waving potential in graphene. *Nature Communications* **5**, 1-6 (2014), Ref. 9–12 in the manuscript). Despite that, only bubbles that move over and encounter the graphene within nanometers (at the same scale of EDL) are capable of generating the fluctuations. Also, the bubble-associated shot noise can be filtered in data analysis by using digital smoothing techniques such as Savitzky-Golay filtering and percentile filtering. Our manuscript focuses on the new flow sensory properties of the graphene. How to tailor and modify the instrumentation for use with real-world complex samples is out of the scope of the manuscript and remains an interesting topic for further investigations. We thank the reviewer for this comment. We have added discussion about the potential bubble-induced interference to our devices and the mitigation strategy in the last paragraph of the Results and Discussion section of the revised manuscript and rearranged this paragraph for clarity.

Reviewers' Comments:

Reviewer #1:

Remarks to the Author:

The authors addressed my comments at a sufficient level.

Reviewer #2:

Remarks to the Author:

The authors have addressed my comments and concerns. I have no further comments.